# Structural basis for the promiscuous PAM recognition by *Corynebacterium diphtheriae* Cas9

Seiichi Hirano[1], Omar O. Abudayyeh[2,3], Jonathan S. Gootenberg[2,3], Takuro Horii[4], Ryuichiro Ishitani[1], Izuho Hatada[4], Feng Zhang[2,3,5,6], Hiroshi Nishimasu[1] & Osamu Nureki[1]

The RNA-guided DNA endonuclease Cas9 cleaves double-stranded DNA targets bearing a protospacer adjacent motif (PAM) and complementarity to an RNA guide. Unlike other Cas9 orthologs, *Corynebacterium diphtheriae* Cas9 (CdCas9) recognizes the promiscuous NNRHHHY PAM. However, the CdCas9-mediated PAM recognition mechanism remains unknown. Here, we report the crystal structure of CdCas9 in complex with the guide RNA and its target DNA at 2.9 Å resolution. The structure reveals that CdCas9 recognizes the NNRHHHY PAM via a combination of van der Waals interactions and base-specific hydrogen bonds. Moreover, we find that CdCas9 exhibits robust DNA cleavage activity with the optimal 22-nucleotide length guide RNAs. Our findings highlight the mechanistic diversity of the PAM recognition by Cas9 orthologs, and provide a basis for the further engineering of the CRISPR-Cas9 genome-editor nucleases.

[1] Department of Biological Sciences, Graduate School of Science, The University of Tokyo, 7-3-1 Hongo, Bunkyo-ku, Tokyo 113-0033, Japan. [2] Broad Institute of MIT and Harvard, Cambridge, MA 02142, USA. [3] McGovern Institute for Brain Research, Massachusetts Institute of Technology, Cambridge, MA 02139, USA. [4] Laboratory of Genome Science, Biosignal Genome Resource Center, Institute for Molecular and Cellular Regulation, Gunma University, 3-39-15 Showa-machi, Maebashi, Gunma 371-8512, Japan. [5] Department of Biological Engineering, Massachusetts Institute of Technology, Cambridge, MA 02139, USA. [6] Department of Brain and Cognitive Sciences, Massachusetts Institute of Technology, Cambridge, MA 02139, USA. Correspondence and requests for materials should be addressed to H.N. (email: nisimasu@bs.s.u-tokyo.ac.jp) or to O.N. (email: nureki@bs.s.u-tokyo.ac.jp)

The RNA-guided DNA endonuclease Cas9, from the type II CRISPR (clustered regularly interspaced short palindromic repeat)-Cas (CRISPR-associated) system, binds CRISPR RNA and trans-activating crRNA or a synthetic single-guide RNA (sgRNA), and cleaves double-stranded DNA targets complementary to the crRNA guide[1–4]. Besides the crRNA-target DNA complementarity, DNA recognition by Cas9 requires a protospacer adjacent motif (PAM), a specific DNA sequence located downstream of the target sequence[5,6]. Since the two-component system, consisting of Streptococcus pyogenes Cas9 (SpCas9) and its sgRNA, can target endogenous genomic sites in a wide range of cell types and organisms, the CRISPR-Cas9 system has been harnessed for numerous technologies, such as genome editing, transcriptional regulation, and epigenetic modulation[7]. Cas9 orthologs, including Staphylococcus aureus Cas9 (SaCas9)[8] and Campylobacter jejuni Cas9 (CjCas9)[9,10], recognize distinct guide RNAs and PAMs. SpCas9, SaCas9, and CjCas9 recognize NGG (N is any nucleotide), NNGRRT (R is A or G), and NNNVRYAC (V is A, G, or C; Y is T or C) as the PAMs, respectively[8–11]. Thus, the use of Cas9 orthologs expands the target space in Cas9-mediated genome editing, and enables the simultaneous targeting of multiple sites in an orthogonal manner[12].

Previous structural studies of SpCas9 provided mechanistic insights into the RNA-guided DNA cleavage by Cas9[13–17]. SpCas9 adopts a bilobed architecture, comprising recognition (REC) and nuclease (NUC) lobes, and accommodates the guide RNA-target DNA heteroduplex in a central channel between the two lobes. The REC lobe mainly consists of α helices and recognizes the RNA–DNA heteroduplex and the sgRNA scaffold. The NUC lobe consists of the RuvC, HNH, Wedge (WED), and PAM-interacting (PI) domains. The PAM-containing DNA duplex is bound between the WED and PI domains, where PAM nucleotides are recognized by a specific set of amino-acid residues in the PI domain. The PAM recognition facilitates the unwinding of the double-stranded DNA target, thereby triggering the base pairing between the crRNA guide and the DNA target. The HNH domain cleaves the DNA strand complementary to the crRNA guide (the target strand), while the RuvC domain cleaves the non-complementary strand (the non-target strand). The crystal structures of several Cas9 orthologs revealed the conserved RNA-guided DNA targeting mechanism, and illuminated the mechanistic diversity of the sgRNA and PAM recognition[9,18,19].

A previous study showed that, unlike other Cas9 orthologs, Corynebacterium diphtheriae Cas9 (CdCas9) recognizes NNRHHHY (H is A, T, or C) as the PAM[8]. Since CdCas9 can recognize a variety of PAM sequences, including the G-less NNAAAAY, the use of CdCas9 could potentially contribute to extending the target range in Cas9-mediated genome editing. However, CdCas9 exhibited slower DNA cleavage kinetics in vitro[20], and failed to induce indels at endogenous target sites in human cells[8]. While the Cas9 orthologs require different guide lengths for efficient DNA cleavage (20-, 21-, and 22-nt guides are optimal for SpCas9, SaCas9, and CjCas9, respectively)[8,10], the cleavage activity of CdCas9 has been examined with only the 20-nt guide sgRNA[8,20]. Thus, it is possible that CdCas9 would exhibit more robust activity with the optimal sgRNA. In addition, the preference of CdCas9 for the 108 possible PAM sequences with the NNRHHHY consensus remains elusive. The PAM recognition mechanism of CdCas9 is unknown, due to the lack of structural information about CdCas9 and the limited sequence similarity between CdCas9 and the other structurally characterized Cas9 orthologs.

Here, we performed functional and structural characterizations of CdCas9. We confirmed that CdCas9 recognizes the NNRHHHY PAM, and found that CdCas9 efficiently cleaves the double-stranded DNA target, when programmed with the 22-nt guide sgRNA. Furthermore, we determined the crystal structure of CdCas9 in complex with the sgRNA and its target DNA, and obtained insights into its promiscuous PAM recognition mechanism. Our findings enhance our mechanistic understanding of the diverse CRISPR-Cas9 nucleases.

## Results

**Biochemical characterization of CdCas9.** To examine the optimal guide length for CdCas9, we performed in vitro cleavage experiments, using the purified CdCas9, the sgRNAs containing 20–24-nt guide sequences (sgRNA20–sgRNA24), and the linearized plasmid DNA containing the 24-nt target sequence and the GGGAAAC PAM. Consistent with previous studies[8,20], CdCas9 with sgRNA20 did not cleave the target DNA efficiently (Fig. 1a, Supplementary Fig. 1a). In contrast, sgRNA21–sgRNA24 facilitated CdCas9-mediated DNA cleavage, with sgRNA22 being optimal (Fig. 1a, Supplementary Fig. 1a). CdCas9-sgRNA22 almost completely cleaved the target DNA in 2 min under our conditions, but its cleavage kinetics was slower than that of SpCas9 (Fig. 1b, Supplementary Fig. 1b). Recent studies showed that the RNA–DNA base pairing at the PAM-distal region (i.e., 20-bp RNA–DNA heteroduplex formation) is important for the activation of the HNH nuclease domain in SpCas9[21,22], suggesting that CdCas9 requires the 22-bp, rather than 20-bp, RNA–DNA heteroduplex formation for the HNH activation. To examine the effect of the guide length on the HNH activation in CdCas9, we measured the target strand cleavage by the HNH domain, using the CdCas9 RuvC-inactive D10A mutant, sgRNA (sgRNA20 or sgRNA22), and a circular plasmid DNA target. The target DNA was nicked more efficiently by the CdCas9 D10A mutant with the sgRNA22, as compared to the sgRNA20 (Supplementary Fig. 1c), suggesting that CdCas9 requires the 22-bp RNA–DNA heteroduplex formation for the HNH activation. These results revealed that CdCas9 can cleave the double-stranded DNA target, when programmed with the optimal sgRNAs.

We next performed a PAM identification assay, using the purified CdCas9-sgRNA complex and a PAM library. We confirmed that CdCas9 recognizes the NNRHHHY PAM (Fig. 1c, Supplementary Fig. 2a), consistent with a previous report in which the PAM library was cleaved by a lysate prepared from CdCas9-expressing human cells[8]. To further examine the PAM preference, we compared the in vitro cleavage activities of CdCas9-sgRNA22 toward 22 plasmid targets, in which each nucleotide in the optimal GGGAAAC PAM was individually substituted with four possible nucleotides. CdCas9 efficiently cleaved the target plasmids with the NGGAAAC and GNGAAAC PAMs (Fig. 1d, Supplementary Fig. 2b), confirming that CdCas9 has no preference for the first and second PAM nucleotides. CdCas9 efficiently cleaved the GGRAAAC targets, but not the GGYAAAC targets (Fig. 1d, Supplementary Fig. 2b), indicating the requirement of the third R for the PAM recognition. CdCas9 showed higher activities for the GGGHAAC targets than the GGGGAAC target (Fig. 1d, Supplementary Fig. 2b), indicating the preference for the fourth H. CdCas9 showed the A > C > T > G preference at positions 5 and 6 in the NNRHHHY PAM (Fig. 1d, Supplementary Fig. 2b). CdCas9 was more active at the GGGAAAY targets than the GGGAAAR targets (Fig. 1d, Supplementary Fig. 2b), indicating the preference for the seventh Y. To further investigate the preference at positions 4–6, we compared the cleavage activities towards the four GGGNNNC targets. CdCas9 was much more active toward the GGGAAAC target relative to the GGGTTTC and GGGCCCC targets, and failed to cleave the GGGGGGC target (Fig. 1d, Supplementary

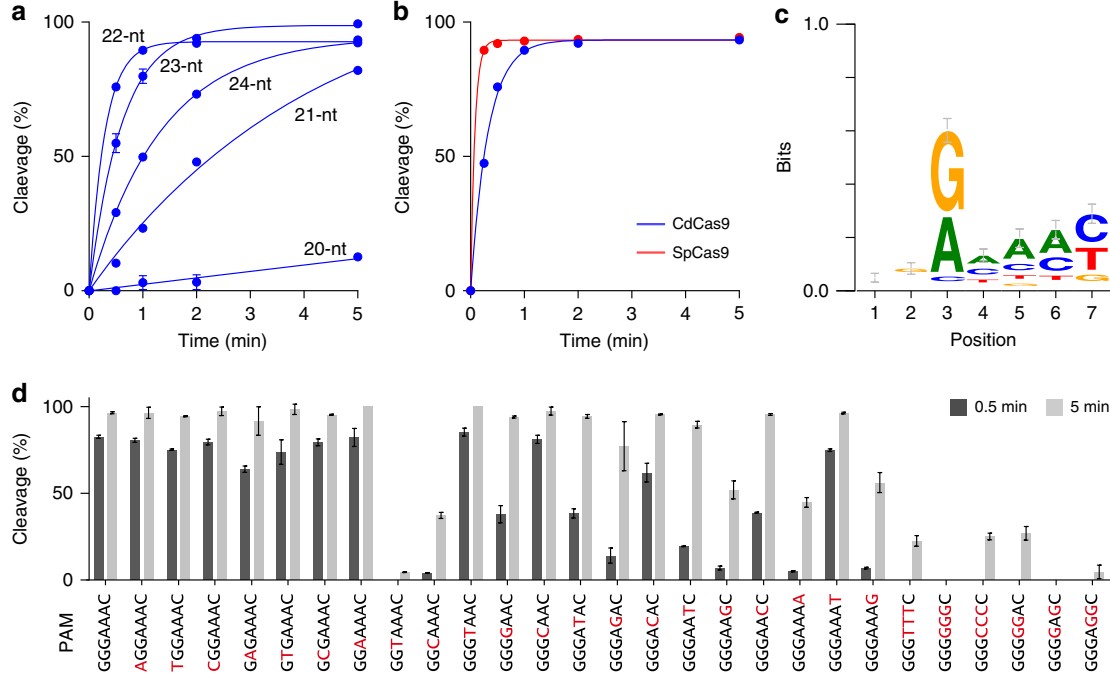

**Fig. 1** In vitro cleavage activities of *Corynebacterium diphtheriae* Cas9 (CdCas9). **a** In vitro cleavage activities of CdCas9 with the 20–24-nt guide single-guide RNAs (sgRNAs). The linearized plasmid target bearing the GGGAAAC protospacer adjacent motif (PAM) was incubated with the CdCas9-sgRNA complex at 37 °C for 0.5, 1, 2, and 5 min, and the cleavage products were then analyzed by a MultiNA microchip electrophoresis system. **b** In vitro cleavage activities of CdCas9 and *Streptococcus pyogenes* Cas9 (SpCas9). CdCas9 and SpCas9 were programmed with their 22- and 20-nt guide sgRNAs, respectively. The linearized plasmid target bearing the GGGAAAC PAM was incubated with the Cas9-sgRNA complex at 37 °C for 0.25, 0.5, 1, 2, and 5 min. **c** Motif obtained from the in vitro PAM identification assay. **d** PAM preference of CdCas9. The linearized plasmid targets bearing the different PAMs were incubated with CdCas9-sgRNA22 at 37 °C for 0.5 and 5 min. Error bars represent s.d. from $n = 3$ replicates

Fig. 2b). In addition, CdCas9 showed almost no activities toward the GGGGGAC, GGGGAGC, and GGGAGGC targets (Fig. 1d, Supplementary Fig. 2b). These results revealed that CdCas9 prefers adenines and rejects guanines at positions 4–6 in the NNRHHHY PAM. We thus concluded that, unlike the other Cas9 orthologs, CdCas9 recognizes the promiscuous NNRHHHY PAM, with a preference for adenine at the fourth to sixth PAM nucleotides.

**CdCas9-mediated genome editing in human cells.** A previous study showed that the vector-expressed CdCas9-sgRNA20 fails to induce indels in human cells[8]. Since CdCas9 requires the 22-nt guide length for robust DNA cleavage in vitro (Fig. 1a), we examined whether CdCas9 induces indels in human cells, using sgRNA20, sgRNA22, or sgRNA24 targeting 16 sites in the *DNMT1*, *DYRK1A*, or *EMX1* locus (Supplementary Table 1). In contrast to our in vitro data, CdCas9 with sgRNA20–sgRNA24 failed to edit these target sites (Supplementary Table 1). Next, we microinjected the CdCas9-sgRNA (sgRNA20, sgRNA22, or sgRNA24) ribonucleoprotein (RNP) complexes, targeting eight sites in the *Tet1EX4*, *Tet1EX7*, or *Tet1EX12* locus, into mouse zygotes (Supplementary Table 2). The CdCas9-sgRNA20 RNPs did not induce indels at the target sites (Supplementary Fig. 3a). In contrast, the CdCas9-sgRNA22 RNPs edited *Tet1EX4* (5%, 1 out of 21 embryos) and *Tet1EX12* (8%, 2 out of 21 embryos), while the CdCas9-sgRNA24 RNPs edited *Tet1EX12* (56%, 10 out of 18 embryos) (Supplementary Fig. 3a). We confirmed that CdCas9-sgRNA22 cleaves the DNA targets with the GTATAAT (*Tet1EX4*) and TGGTAAT (*Tet1EX12*) PAMs in vitro (Supplementary Fig. 3b, c), excluding the possibility that the inefficient editing at these sites was due to the inappropriate PAM

sequences. These results revealed that the CdCas9-sgRNA RNPs can be used for genome editing in mammalian cells, albeit with low efficiency.

**Crystal structure of the CdCas9-sgRNA-target DNA complex.** To elucidate the CdCas9-mediated DNA cleavage mechanism, we attempted to determine the crystal structure of CdCas9 (1084 residues) in complex with an sgRNA and its target DNA. Since we failed to obtain diffraction-quality crystals, we crystallized a CdCas9-ΔHNH variant, in which the HNH domain (residues 498–663) is replaced by a GGGSGG linker, as in the case of CjCas9[9] (Fig. 2a). We determined the crystal structure of CdCas9-ΔHNH in complex with a 112-nt sgRNA (a 20-nt guide sequence), a 28-nt target DNA strand, and an 8-nt non-target DNA strand (the GGGTAAT PAM), at 2.9 Å resolution (Fig. 2b, c, Table 1). The crystal structure revealed that CdCas9 adopts a bilobed architecture consisting of the REC and NUC lobes, with the guide RNA-target DNA heteroduplex bound within the central channel, as in the other Cas9 structures (Fig. 2c). CdCas9-ΔHNH comprises five domains, including the RuvC (residues 1–51, 449–497, and 664–807), REC1 (residues 86–235), REC2 (residues 236–448), WED (residues 821–904), and PI (residues 905–1084) domains. The RuvC and REC1 domains are connected by an arginine-rich bridge helix (BH) (residues 52–85), while the RuvC and WED domains are connected by a phosphate-lock loop (PLL) (residues 808–820).

The overall structure of CdCas9 is more similar to that of CjCas9 (PDB: 5X2G, r.m.s.d. of 2.7 Å for 496 equivalent Cα atoms) than those of SaCas9 (PDB: 5CZZ, r.m.s.d. of 3.5 Å for 514 equivalent Cα atoms), SpCas9 (PDB: 4UN3, r.m.s.d. of 3.7 Å for 468 equivalent Cα atoms), and *Francisella novicida* Cas9

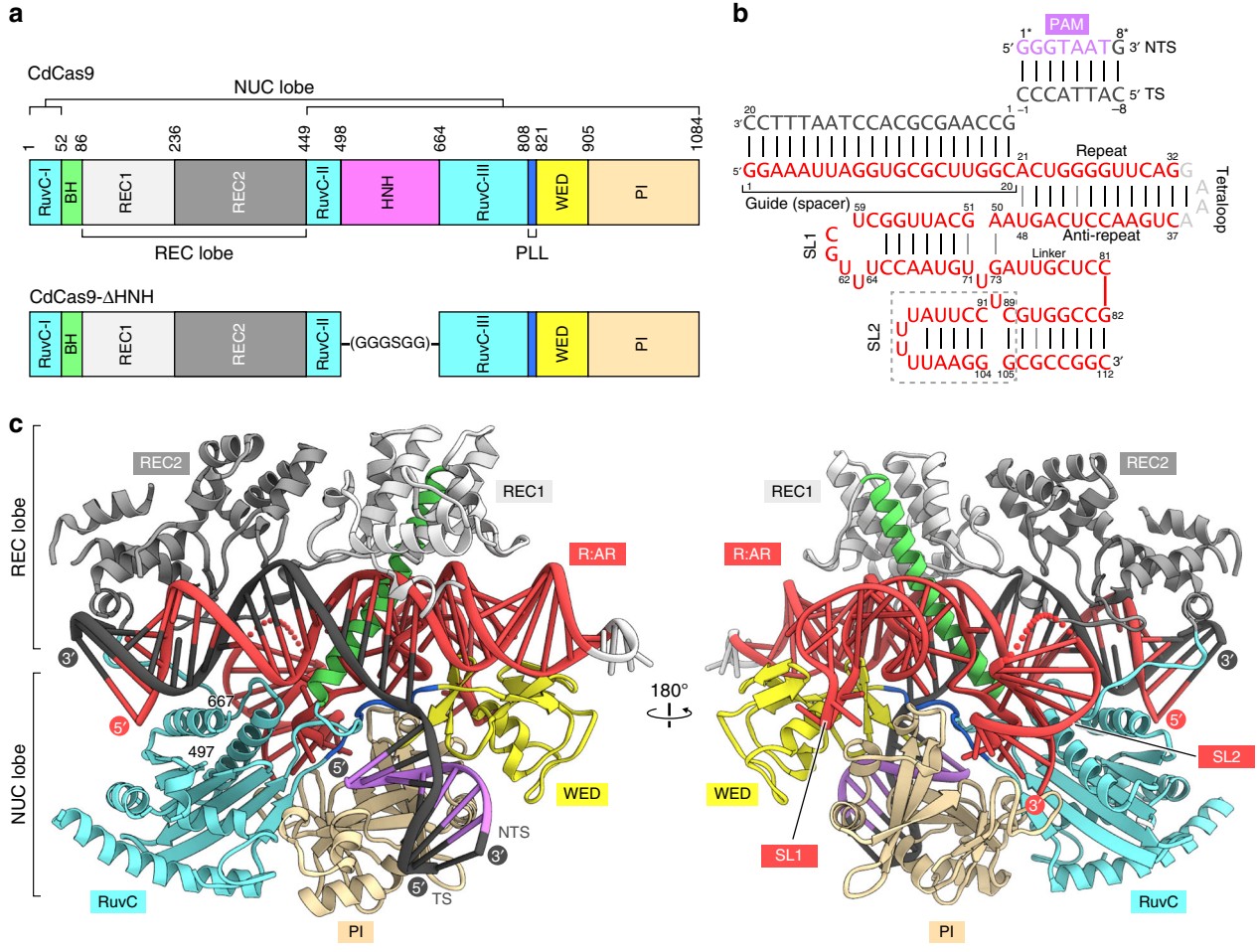

**Fig. 2** Structure of the CdCas9-sgRNA-target DNA complex. **a** Domain structure of CdCas9. The HNH nuclease domain was truncated for crystallization. BH, bridge helix; PLL, phosphate-lock loop. **b** Schematics of the single-guide RNA (sgRNA) and the target DNA. The disordered region is enclosed in a dashed box. TS, target strand; NTS, non-target strand; SL1, stem loop 1; SL2, stem loop 2. **c** Overall structure of CdCas9-ΔHNH in complex with the sgRNA and its target DNA. The BH is colored green. The disordered region of the sgRNA is shown as a dotted line. CdCas9, *Corynebacterium diphtheriae* Cas9; R:AR, repeat:anti-repeat duplex

(FnCas9) (PDB: 5B2O, r.m.s.d. of 4.4 Å for 368 equivalent Cα atoms) (Fig. 3). The RuvC, WED, and PI domains of CdCas9- shares structural similarity with those of *Actinomyces naeslundii* Cas9 (PDB: 4OGE, r.m.s.d. of 1.3 Å for 441 equivalent Cα atoms)[13] (Fig. 3).

The sgRNA guide segment (G1–C20) and the target DNA strand (dG1–dC20) form the RNA–DNA heteroduplex, which is accommodated in the central channel (Fig. 2c). The target DNA strand (dC(−1)–dC(−8)) and the non-target DNA strand (dG1*–dG8*) form the PAM-containing DNA duplex, which is bound between the WED and PI domains (Fig. 2c). As in the other Cas9 structures, the sgRNA "seed" region (C13–C20) is extensively recognized by the BH and the REC1 domain, while the backbone phosphate group between dG1 and dC(−1) in the target DNA strand is recognized by the PLL (Fig. 2c). These conserved structural features indicated that the RNA-guided DNA cleavage mechanism of CdCas9 is similar to those of the other Cas9 orthologs.

**Structure and recognition of the sgRNA scaffold.** The sgRNA consists of the guide segment (G1–C20), the repeat:anti-repeat duplex (A21:U48–G32:C37), the tetraloop (G33–A36), the stem loop 1 (A50–G73), the single-stranded linker (A74–C81), and the

stem loop 2 (G82–C112) (Figs. 4, 5a). As expected from the nucleotide sequence, the repeat:anti-repeat duplex adopts an A-form-like conformation, and is recognized by the BH and the REC1/WED domains (Figs. 4, 5b). Stem loop 1 is recognized by the BH, the PLL, and the WED/PI domains (Figs. 4, 5b). The deletion of nucleotides 57–65 reduced the CdCas9-mediated DNA cleavage (Fig. 5c, Supplementary Fig. 4a), indicating the functional importance of stem loop 1. The basal region of stem loop 2 is recognized by the RuvC and PI domains (Figs 4, 5b), while the upper region of stem loop 2 (C89–G105) is disordered in the crystal structure (Supplementary Fig. 4b). Indeed, the deletion of nucleotides 82–112, but not nucleotides 87–107, reduced the DNA cleavage activity of CdCas9 (Fig. 5c, Supple- mentary Fig. 4a). The linker region is recognized by the BH, the PLL, and the RuvC/PI domains in a base-specific manner (Figs. 4, 5d). In particular, G77, C78, U79, and C80/C81 form hydrogen bonds with Asp977, Arg1070, Asp939, and His1076, respectively (Fig. 5d). The present structure revealed that the CdCas9-sgRNA adopts a conformation distinct from that of the SpCas9 sgRNA, consistent with their different nucleotide sequences (Fig. 5e). Nonetheless, a previous study reported that the SpCas9 sgRNA can support DNA cleavage by CdCas9[20]. We thus examined the ability of the SpCas9 sgRNA to support the CdCas9-mediated DNA cleavage, and found that CdCas9 with the SpCas9 sgRNA

**Table 1 Data collection and refinement statistics**

| | |
|---|---|
| **Data collection** | |
| Beamline | SPring-8 BL41XU |
| Wavelength (Å) | 0.9790 |
| Space group | C2 |
| Cell dimensions | |
| $a, b, c$ (Å) | 139.0, 119.0, 116.3 |
| $\alpha, \beta, \gamma$ (°) | 90, 113.6, 90 |
| Resolution (Å)[a] | 106.6-2.9 (3.03-2.90) |
| $R_{merge}$ | 0.168 (3.024) |
| $R_{pim}$ | 0.047 (0.844) |
| $I/\sigma I$ | 10.2 (1.4) |
| Completeness (%) | 100.0 (100.0) |
| Multiplicity | 13.4 (13.5) |
| CC(1/2) | 0.999 (0.802) |
| **Refinement** | |
| Resolution (Å) | 67.9-2.9 (3.00-2.90) |
| No. of reflections | 38,462 (3795) |
| $R_{work}/R_{free}$ | 0.221/0.254 (0.399/0.469) |
| No. of atoms | |
| Protein | 6292 |
| Nucleic acid | 2755 |
| Ion | 1 |
| Solvent | 11 |
| $B$-factors (Å$^2$) | |
| Protein | 116.1 |
| Nucleic acid | 112.0 |
| Ion | 121.5 |
| Solvent | 72.8 |
| R.m.s. deviations | |
| Bond lengths (Å) | 0.003 |
| Bond angles (°) | 0.54 |
| Ramachandran plot (%) | |
| Favored region | 96.94 |
| Allowed region | 2.94 |
| Outlier region | 0.12 |
| MolProbity score | |
| Clashscore | 6.44 |
| Rotamer outlier | 5.00 |

[a]Values within parentheses are for the highest resolution shell

does not cleave the target DNA in vitro (Fig. 5c, Supplementary Fig. 4a). Together, these observations demonstrated that CdCas9 specifically recognizes its cognate guide RNA in a manner distinct from those of the other Cas9 orthologs.

**Recognition of the NNRHHHY PAM.** In the present structure, the PAM duplex is bound between the WED and PI domains, where the GGGTAAT PAM is recognized by multiple residues in the PI domain (Fig. 6a, b). The dG1* nucleobase does not directly contact the protein (Fig. 6c), consistent with the lack of a preference for the first PAM nucleotide. Unexpectedly, the O6 and N7 of dG2* form bidentate hydrogen bonds with Arg1042 (Fig. 6c), despite the lack of an observed preference for the second PAM nucleotide (Fig. 1c, d). The R1042A mutant showed slightly reduced DNA cleavage activity (Fig. 6d, Supplementary Fig. 5a), indicating that Arg1042 is involved in the PAM recognition. To explore the importance of the second PAM nucleotide, we compared the in vitro cleavage activities of the wild-type CdCas9 towards the GNGGAAC, GNGAGAC, and GNGAAGC targets. CdCas9 cleaved the GGGGAAC/GGGAGAC/GGGAAGC targets more efficiently, as compared to the GHGGAAC/GHGAGAC/GHGAAGC targets (Fig. 6e, Supplementary Fig. 5b), suggesting the functional importance of the interaction between Arg1042 and the second G nucleotide for the recognition of the suboptimal

PAMs. The N7 of dG3* forms a hydrogen bond with Arg1017 (Fig. 6c), and the R1017A mutant showed almost no activity (Fig. 6d, Supplementary Fig. 5a), confirming the importance of Arg1017 for the PAM recognition. Since the N7 is common in the purines, this interaction can explain the requirement for the third R in the PAM. While the nucleobases of dT4*–dA6* in the non-target strand do not form direct contacts with the protein, the nucleobases of dA(−4)–dT(−6) in the target strand are located in the vicinity of a hydrophobic patch formed by Phe1011, Lys1015, Pro1043, and Leu1046 (Fig. 6c). The single mutants (F1011A, K1015A, P1043A, and L1046A) showed reduced DNA cleavage activities, and the triple (F1011A/P1043A/L1046A) and quadruple (F1011A/K1015A/P1043A/L1046A) mutants showed almost no cleavage activities (Fig. 6d, Supplementary Fig. 5a), confirming the functional importance of the hydrophobic patch for the PAM recognition. Molecular modeling suggested that the methyl groups of the fourth to sixth T nucleotides in the target strand form van der Waals interactions with the hydrophobic patch (Supplementary Fig. 6), consistent with the preference for the A nucleotides at the fourth to sixth PAM positions. In contrast, the 4-amino group of the C nucleotides at these positions sterically clashed with the hydrophobic patch (Supplementary Fig. 6), explaining why CdCas9 disfavors the G nucleotides at positions 4–6 in the NNRHHHY PAM. The nucleobase of dT7* does not directly contact the protein, while the N7 of dA(−7) forms a hydrogen bond with Lys1015 (Fig. 6c). The interaction between Lys1015 and the seventh R in the target strand can explain the preference of CdCas9 for the seventh Y in the PAM. Together, these structural observations revealed that CdCas9 recognizes the promiscuous NNRHHHY PAM, via a combination of hydrogen bonds and van der Waals interactions with both the target and non-target strands in the PAM duplex.

## Discussion

In the CRISPR-Cas immune system, the PAM plays central roles in the self versus non-self discrimination[23]. Cas9 selectively targets protospacer sequences with the PAM in foreign DNAs but not spacer sequences in the host CRISPR array, due to the absence of the PAM in the spacer-flanking repeat sequences. We showed that, unlike the other Cas9 orthologs, CdCas9 recognizes the promiscuous NNRHHHY PAM, raising the question of how the CRISPR array in *C. diphtheriae* escapes from self-targeting. Intriguingly, the spacer-flanking repeat sequence (the 5′ region of the guide RNA scaffold) is ACTGGGG, which does not match the NNRHHHY PAM (Fig. 2b). Thus, CdCas9 recognizes the promiscuous PAM, but avoids self-targeting in the CRISPR-Cas immune system.

A structural comparison of CdCas9 with the other Cas9 orthologs revealed that their PI domains have limited sequence similarity (Fig. 7a), but share a conserved core fold (consisting of a three-stranded anti-parallel β-sheet (β1–β3) and a four-stranded anti-parallel β-sheet (β4–β7)), in which distinct amino-acid residues in the β5–β7 region participate in the PAM recognition (Fig. 7b–f). In SpCas9 and SaCas9, the arginine residues in the β7 region (Arg1333/Arg1335 of SpCas9 and Arg1015 of SaCas9) form bidentate hydrogen-bonding interactions with the G nucleotides in their PAMs[15,18] (Fig. 7b, c). In contrast, in CdCas9, Lys1015, and Arg1017 in β6 form single hydrogen bonds with the R nucleotides, and the hydrophobic residues in β5 and β7 provide favorable interactions with the T nucleotides, thereby enabling the promiscuous PAM recognition (Fig. 7d). In addition, the present structure revealed that, whereas SpCas9, SaCas9, and FnCas9 mainly recognize their PAM nucleotides in the non-target strand, CdCas9 recognizes the nucleotides in both the target and non-target strands, as observed

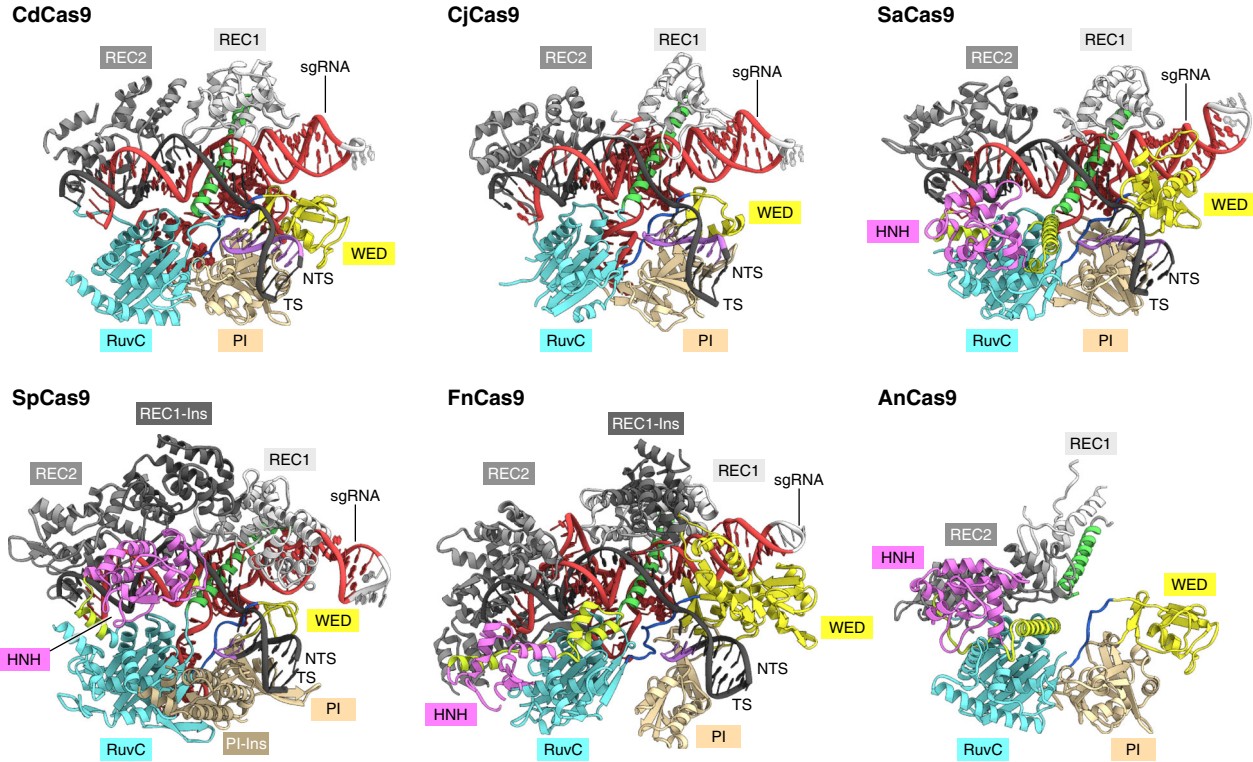

**Fig. 3** Structural comparison of the Cas9 orthologs. The structures of *Corynebacterium diphtheriae* Cas9 (CdCas9), *Campylobacter jejuni* Cas9 (CjCas9) (PDB: 5X2G), *Staphylococcus aureus* Cas9 (SaCas9) (PDB: 5CZZ), *Streptococcus pyogenes* Cas9 (SpCas9) (PDB: 4UN3), and *Francisella novicida* Cas9 (FnCas9) (PDB: 5B2O) bound to their cognate single-guide RNA (sgRNAs) and target DNAs, and the structure of *Actinomyces naeslundii* Cas9 (AnCas9) in the apo form (PDB: 4OGE). The bridge helix and the linker regions are colored green and light green, respectively

in CjCas9[9] (Fig. 7d, e). These structural observations highlight the mechanistic diversity in the PAM recognition by the Cas9 orthologs.

Previous studies showed that SpCas9, SaCas9, and CjCas9 require 20-, 21-, and 22-nt guides for efficient genome editing, respectively[8,10]. Our biochemical data revealed that CdCas9 requires a 22-nt guide for robust DNA cleavage, reinforcing the notion that the optimal guide lengths are different among the Cas9 orthologs. Recent studies have shown that the RNA–DNA base pairing at the PAM-distal region is important for the HNH activation, and is monitored by the REC2 domain in SpCas9[21,22]. Notably, a structural comparison between the Cas9 orthologs revealed the conformational differences in their REC2 domains (Supplementary Fig. 7), suggesting the differences in their RNA–DNA sensing mechanisms, consistent with their different optimal guide lengths.

In summary, our structural and functional data highlight the mechanistic diversity of the Cas9 enzymes, and provide a basis for the improvement of the utility of the CRISPR-Cas9 technology. CdCas9 recognizes G-less PAM sequences, such as NNAAAAY, whereas most Cas9 orthologs require G-rich PAMs. Nonetheless, in contrast to our in vitro data, CdCas9 lacked robust activity in mammalian cells. One of the reasons may be the inefficient formation of the CdCas9-sgRNA RNPs in mammalian cells, given that CdCas9 binds its sgRNA less tightly as compared to SpCas9[20]. If so, protein engineering of CdCas9 to enhance the sgRNA binding may improve the efficiency of CdCas9-mediated genome editing.

## Methods

**Sample preparation**. The gene encoding full-length CdCas9 (residues 1–1084) was codon optimized, synthesized (GenScript), and cloned between the *Nde*I and *Xho*I sites of the modified pE-SUMO vector (LifeSensors) (Supplementary Tables 3, 4).

For crystallization, we prepared the CdCas9-ΔHNH variant lacking the HNH domain (residues 498–663), in which His497 (RuvC-II) and Ser664 (RuvC-III) are connected by a GGGSGG linker (Supplementary Table 3). CdCas9-ΔHNH was created by a PCR-based method, using the vector encoding the full-length CdCas9 as the template (Supplementary Table 5). CdCas9-ΔHNH was expressed at 20 °C in *Escherichia coli* Rosetta 2 (DE3) (Novagen). The *E. coli* cells were cultured at 37 °C in LB medium (containing 20 mg/L kanamycin) until the $OD_{600}$ reached 0.8, and then protein expression was induced by the addition of 0.1 mM isopropyl-β-D-thiogalactopyranoside (Nacalai Tesque) and an incubation at 20 °C for 20 h. The *E. coli* cells were resuspended in buffer A (50 mM Tris-HCl, pH 8.0, 20 mM imidazole, and 1 M NaCl), lysed by sonication, and then centrifuged. The supernatant was mixed with Ni-NTA Superflow (Qiagen). The protein was eluted with buffer B (50 mM Tris-HCl, pH 8.0, 0.3 M imidazole, and 0.3 M NaCl). The protein was loaded onto a HiTrap Heparin HP column (GE Healthcare) equilibrated with buffer C (20 mM Tris-HCl, pH 8.0, and 0.3 M NaCl). The protein was eluted with a linear gradient of 0.3–2 M NaCl. To remove the His₆-SUMO-tag, the purified protein was mixed with TEV protease, and was dialyzed at 4 °C overnight against buffer D (20 mM Tris-HCl, pH 8.0, 40 mM imidazole, and 0.5 M NaCl). The protein was passed through the Ni-NTA column equilibrated with buffer D. The protein was further purified by chromatography on a HiLoad Superdex 200 16/60 column (GE Healthcare) equilibrated with buffer E (10 mM Tris-HCl, pH 8.0, and 150 mM NaCl). The selenomethionine (SeMet)-substituted CdCas9-ΔHNH was expressed in *E. coli* B834 (DE3) (Novagen), and was purified using a similar protocol to that for the native protein. The sgRNA was transcribed in vitro with T7 RNA polymerase, using a PCR-amplified DNA template. The transcribed RNA was purified by 8% denaturing (7 M urea) polyacrylamide gel electrophoresis. The target and non-target DNA strands were purchased from Sigma-Aldrich. The purified CdCas9-ΔHNH protein was mixed with the sgRNA, the target DNA strand, and the non-target DNA strand (the GGGTAAT PAM) (molar ratio, 1:1.5:2.3:2.7), and then the CdCas9-sgRNA-DNA complex was purified by gel filtration chromatography on a Superdex 200 Increase 10/300 column (GE Healthcare) equilibrated with buffer E. For in vitro cleavage assays, the mutants of CdCas9 were created by a PCR-based method, using the vector encoding the full-length CdCas9 as the template (Supplementary Table 5). The wild-type and mutants of full-length CdCas9 were expressed and purified, using a protocol similar to that for CdCas9-ΔHNH. All of the sgRNAs used for in vitro cleavage assays were transcribed in vitro, and then were purified using an RNeasy Mini Kit (Qiagen).

**Crystallography**. The purified CdCas9-sgRNA-DNA complex was crystallized at 20 °C, using the hanging-drop vapor diffusion method. Crystals were obtained by

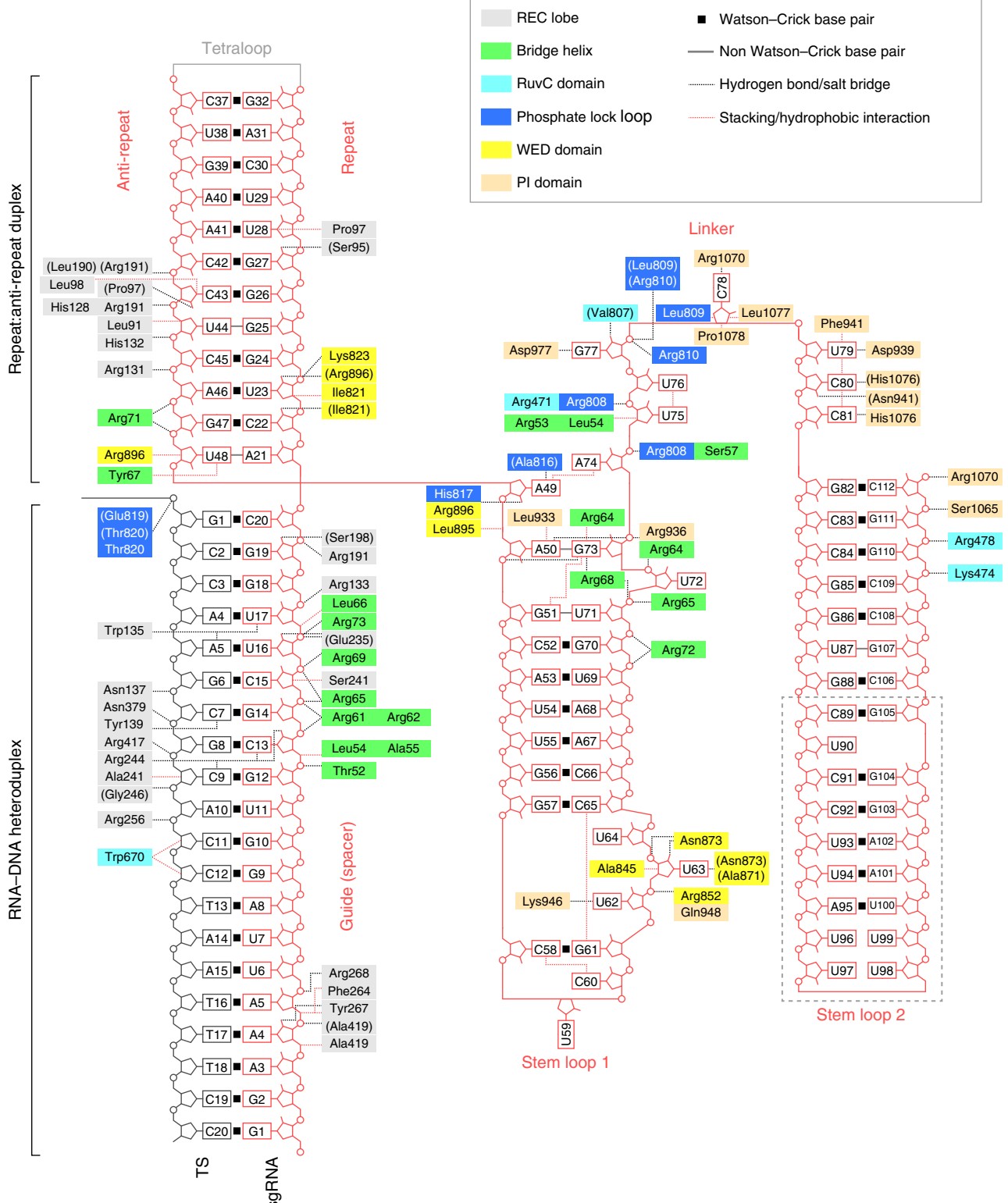

**Fig. 4** Schematic of the nucleic acid recognition by *Corynebacterium diphtheriae* Cas9 (CdCas9). The residues that interact with the single-guide RNA (sgRNA) and its target DNA via their main chains are shown within parentheses. The disordered region is enclosed in a dashed box

mixing 1 μL of complex solution ($A_{260\,nm}$ = 15) and 1 μL of reservoir solution (0.1 M Tris-HCl, pH 8.0, 22–25% PEG 3350, 0.2 M lithium sulfate, and 0.3 M potassium fluoride). The SeMet-labeled complex was crystallized under similar conditions. The crystals were cryoprotected in reservoir solution supplemented with 20% ethylene glycol. X-ray diffraction data were collected at 100 K on beamline BL41XU at SPring-

8 and processed using DIALS[24] and AIMLESS[25]. The structure was determined by the Se-SAD method, using PHENIX AutoSol[26]. The model was automatically built using Buccaneer[27], followed by manual model building using COOT[28] and structural refinement using PHENIX[26]. Data collection statistics are summarized in Table 1. Structural figures were prepared using CueMol (http://www.cuemol.org).

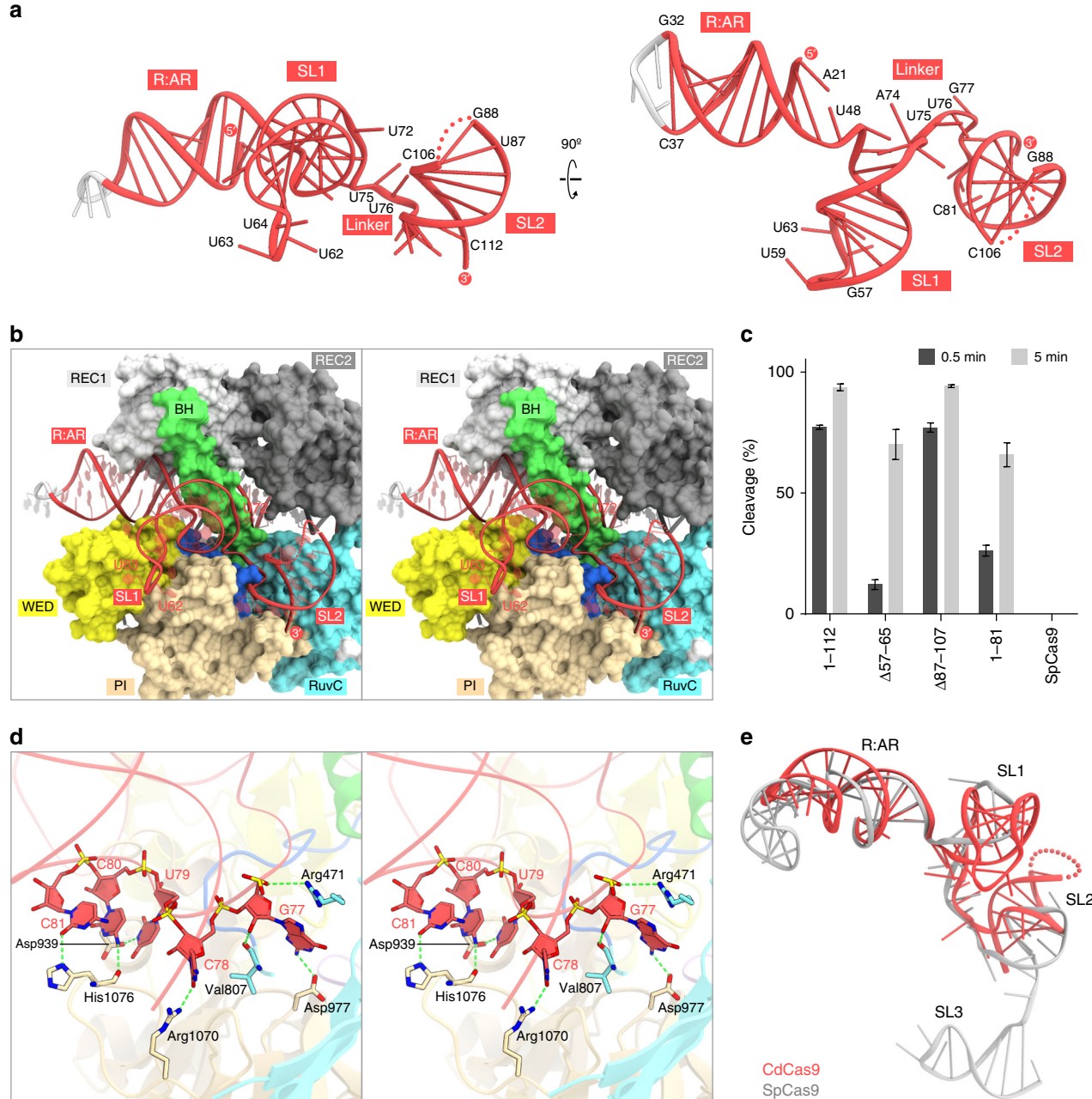

**Fig. 5** Structure and recognition of the single-guide RNA (sgRNA) scaffold. **a** Structure of the sgRNA scaffold. The guide region is omitted for clarity. The disordered region is shown as a dotted line. **b** Binding of the sgRNA scaffold to *Corynebacterium diphtheriae* Cas9 (CdCas9) (stereo view). The disordered region of the sgRNA is shown as a dotted line. **c** Effects of the sgRNA truncation. The linearized plasmid target with the GGGAAAC protospacer adjacent motif (PAM) was incubated at 37 °C for 0.5 and 5 min with CdCas9 and the truncated CdCas9 sgRNA (22-nt guide) or the *Streptococcus pyogenes* Cas9 (SpCas9) sgRNA (22-nt guide). 1–112, the full-length sgRNA; Δ57–65, the sgRNA, in which nucleotides 57–65 were replaced with GAAA; Δ87–107, the sgRNA, in which nucleotides 87–107 were replaced with GAAA; 1–81, the sgRNA, in which nucleotides 82–112 were truncated; SpCas9, the SpCas9 sgRNA. **d** Base-specific recognition of the sgRNA linker region (stereo view). Hydrogen bonds are shown as dashed lines. **e** Superimposition of the CdCas9 sgRNA and the SpCas9 sgRNA (PDB: 4OO8). The disordered region is shown as a dotted line. Error bars represent s.d. from $n = 3$ replicates

**In vitro cleavage assay**. The pUC119 plasmid, containing the 24-nt target sequence and the PAMs, was used as the substrate for in vitro cleavage assays (Supplementary Table 6). The *Eco*RI-linearized pUC119 plasmid (100 ng, 4.7 nM) was incubated at 37 °C for 0.25–30 min with the CdCas9-sgRNA (50 nM) in 10 μL of reaction buffer, containing 20 mM HEPES, pH 7.5, 100 mM KCl, 2 mM MgCl₂, 1 mM dithiothreitol, and 5% glycerol. The reaction was stopped by the addition of quench buffer, containing EDTA (40 mM final concentration) and proteinase K (4 μg). Reaction products were resolved, visualized, and quantified with a MultiNA microchip electrophoresis device (Shimadzu). For the measurement of the cleavage activity of the CdCas9 D10A mutant, the circular pUC119 target plasmid (500 ng, 4.7 nM) was incubated at 37 °C for 0.5–5 min with the CdCas9-sgRNA (50 nM), in 50 μL of the reaction buffer, and the reaction was then stopped by the addition of the quench

buffer. The reaction products were resolved on an ethidium bromide-stained 1% agarose gel, and then visualized using an Amersham Imager 600 (GE Healthcare).

**PAM identification assay**. The PAM identification assay was performed using a PAM library, prepared as previously described[9]. Single-stranded DNA oligonucleotides (Integrated DNA Technologies), containing seven randomized nucleotides downstream of a 20-nt target sequence, were converted to dsDNA via fill-in with the large Klenow fragment (New England Biolabs) and cloned into pUC19 by Gibson cloning (New England Biolabs) to generate a library. The plasmid library was digested in vitro with purified CdCas9 complexed with an sgRNA targeting the PAM library. The cleavage products were resolved on 2% agarose E-gels (Life Technologies), and

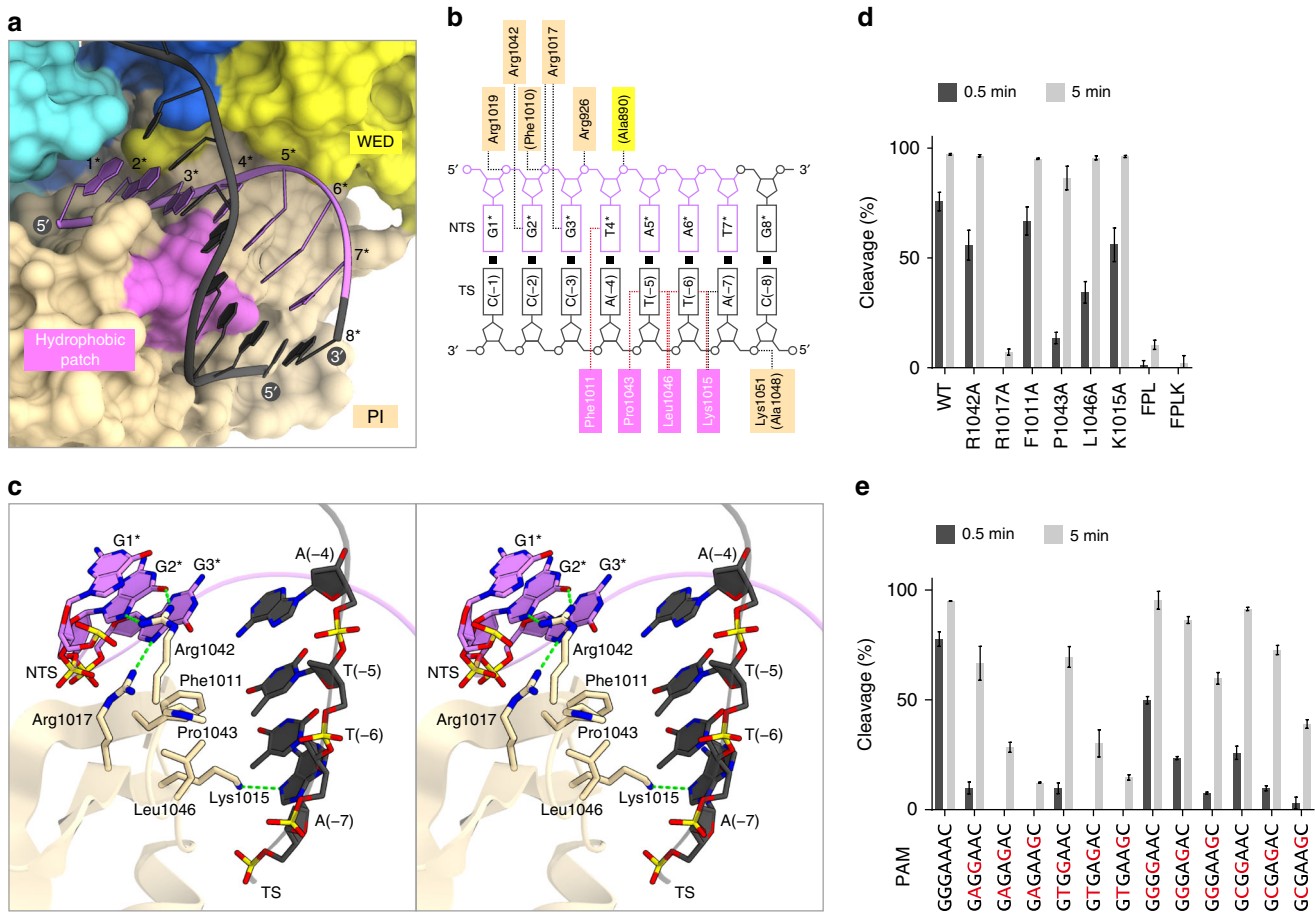

**Fig. 6** Protospacer adjacent motif (PAM) recognition by *Corynebacterium diphtheriae* Cas9 (CdCas9). **a** Binding of the PAM duplex to CdCas9. **b** Schematics of the PAM recognition by CdCas9. Hydrogen bonds and hydrophobic interactions are depicted by black and red dashed lines, respectively. **c** Recognition of the GGGTAAT PAM (stereo view). Hydrogen bonds are shown as dashed lines. **d** Effects of the mutations on the PAM-interacting residues. The linearized plasmid target with the GGGAAAC PAM was incubated with the wild-type or the mutants of CdCas9, together with the single-guide RNA (sgRNA22), at 37 °C for 0.5 and 5 min. FPL, F1011A/P1043A/L1046A; FPLK, F1011A/K1015A/P1043A/L1046A. **e** Functional importance of the second PAM nucleotide. The linearized plasmid targets bearing the different PAMs were incubated with CdCas9-sgRNA22 at 37 °C for 0.5 and 5 min. Error bars represent s.d. from $n = 3$ replicates

the uncleaved target plasmid band was isolated with a Zymoclean Gel DNA Recovery Kit (Zymo Research). Uncleaved PAMs were PCR amplified and sequenced on a MiSeq sequencer (Illumina). The resulting sequence data were analyzed by extracting the seven nucleotide PAM regions, counting the individual PAMs, and normalizing the PAM to the total reads for each sample. For a given PAM sequence, enrichment was calculated as the $\log_2$ ratio compared to a no-protein control, with a 0.01 pseudocount adjustment. PAMs above an enrichment threshold set to 0.3 were used to generate sequence logos[29]. To generate the PAM wheel representation[30], the ratios of PAM abundances as compared to a no-protein control, with a 0.01 pseudocount adjustment, were used directly as the input for Krona[31].

**Indel analysis in human cells**. Gene editing experiments were performed in the human embryonic kidney 293FT (HEK293FT) cell line, which was maintained in Dulbecco's modified Eagle's medium (Gibco) supplemented with 10% fetal bovine serum (FBS) at 37 °C under a 5% $CO_2$ atmosphere. HEK239FT cells were seeded at $2 \times 10^4$ cells per well in 96-well plates, 24 h prior to transfection. Using the Lipofectamine 2000 reagent (Life Technologies), HEK239FT cells were transfected with the plasmid (100 ng) encoding humanized CdCas9 with an N-terminal SV40 nuclear localization tag and the plasmid (50 ng) encoding the U6-driven sgRNAs. Two days post transfection, the genomic DNA was extracted, using 20 μL Quick-Extract DNA Extraction Solution (Epicenter). Insertion/deletion events (indels) were quantified by targeted PCR at the *DNMT1*, *DYRK1A*, or *EMX1* site (Supplementary Table 1), followed by sequencing on a MiSeq sequencer.

**Indel analysis in mouse zygotes**. All animal procedures were approved by the Animal Care and Experimentation Committee at Gunma University and performed in accordance with approved guidelines. Female B6D2F1 mice (8–10 weeks

old, CLEA Japan) were superovulated by the injection of 7.5 units of pregnant mare's serum gonadotropin (ASKA Pharmaceutical), followed by 7.5 units of human chorionic gonadotrophin (hCG; ASKA Pharmaceutical) 48 h later, and then mated overnight with B6D2F1 male mice. Zygotes were collected from oviducts 21 h after the hCG injection, and pronuclei-formed zygotes were placed into the M2 medium. Microinjection was performed using a microscope equipped with a microinjector (Narishige). The CdCas9-sgRNA RNPs were assembled by mixing the purified CdCas9 (40 ng/μL) and the sgRNA (50 ng/μL), targeting the mouse *Tet1EX4*, *Tet1EX7*, or *Tet1EX12* locus (Supplementary Table 2), and then the CdCas9-sgRNA RNPs (1 pL) were injected into the pronuclei of the zygotes. After injection, all zygotes were cultured in the M16 medium for 4 days. To detect indels, the targeted region was amplified by PCR, using the genomic DNA extracted from each blastocyst and the primers (Supplementary Table 7). The PCR products were digested with a specific restriction enzyme that cleaves the Cas9 target site of the unmodified genomes, and then were analyzed by agarose gel electrophoresis. For the *Tet1EX12* target site with the GGGTAAT PAM, indels were detected by a heteroduplex mobility assay. Briefly, the PCR products were reannealed and fractionated by PAGE (polyacrylamide gel electrophoresis) to detect the heteroduplex.

**Quantification and statistical analyses**. In vitro cleavage experiments were performed at least three times. Data are shown as mean ± s.d. ($n = 3$). Kinetics data were fitted with a one-phase exponential association curve, using Prism (GraphPad).

**Reporting summary**. Further information on research design is available in the Nature Research Reporting Summary linked to this article.

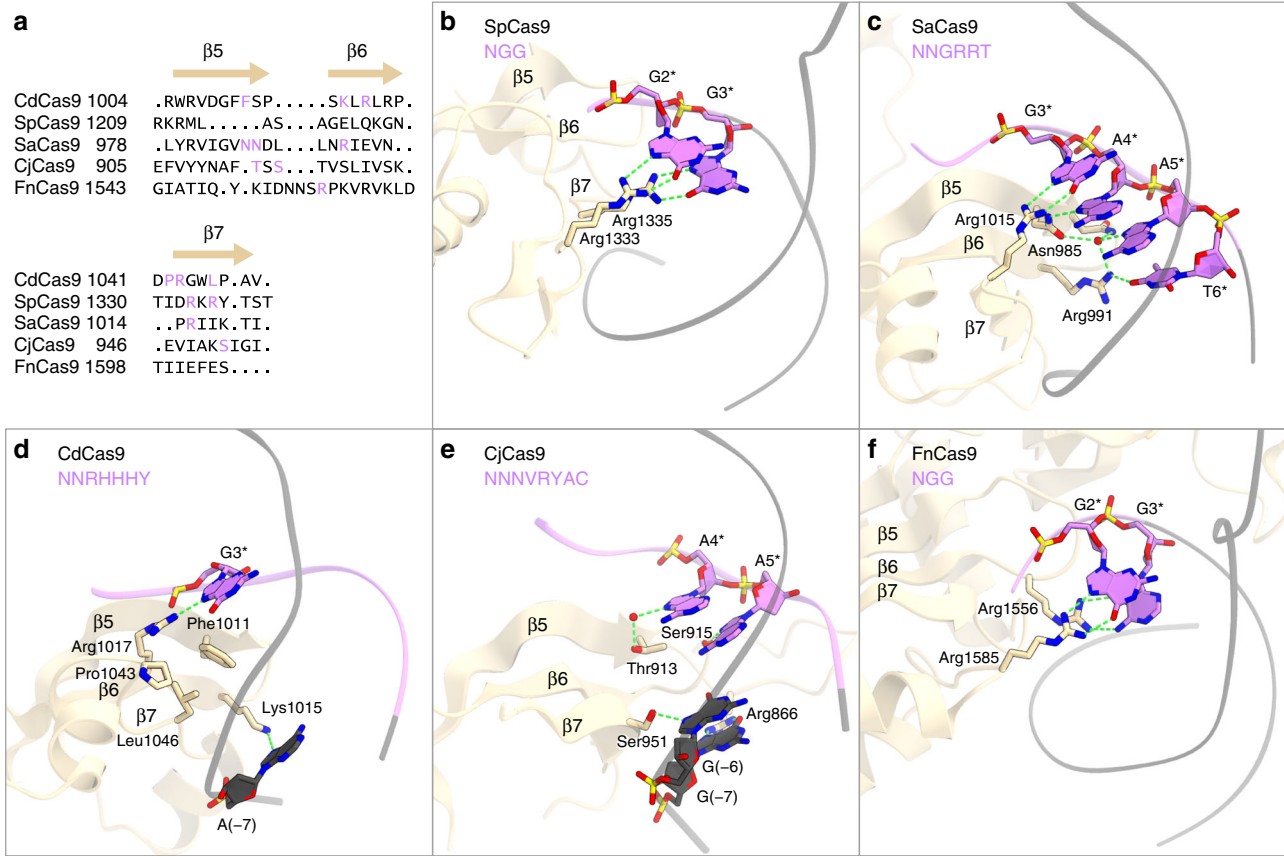

**Fig. 7** Protospacer adjacent motif (PAM) recognition by the Cas9 orthologs. **a** Structure-based sequence alignment of the PAM-interacting regions of the Cas9 orthologs. **b**–**f** PAM recognition by *Streptococcus pyogenes* Cas9 (SpCas9) (PDB: 4UN3) (**b**), *Staphylococcus aureus* Cas9 (SaCas9) (PDB: 5CZZ) (**c**), *Corynebacterium diphtheriae* Cas9 (CdCas9) (**d**), *Campylobacter jejuni* Cas9 (CjCas9) (PDB: 5X2G) (**e**), and *Francisella novicida* Cas9 (FnCas9) (PDB: 5B2O) (**f**). In SpCas9, the insertion between the β6 and β7 is omitted for clarity. The PAMs are highlighted in purple. Hydrogen bonds are shown as dashed lines. Water molecules are depicted as red spheres

## Data availability

The atomic coordinates of the CdCas9-sgRNA-DNA complex have been deposited in the Protein Data Bank, with the accession number PDB: 6JOO. The source data underlying Supplementary Figs. 1c and 3a are provided as a Source Data file. Other data are available from the corresponding authors upon reasonable request.

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

## Acknowledgements

We thank the beamline scientists at BL41XU at SPring-8 for assistance with data collection. J.S.G. is supported by a D.O.E. Computational Science Graduate Fellowship. F.Z. is supported by the National Institutes of Health through NIMH (5DP1-MH100706 and 1R01-MH110049) and NIDDK (5R01DK097768-03), the New York Stem Cell, Simons, Paul G. Allen Family, and Vallee Foundations. F.Z. is a New York Stem Cell Foundation Robertson Investigator. F.Z. is a founder of Editas Medicine and a scientific advisor for Editas Medicine and Horizon Discovery. H.N. is supported by JST, PRESTO, JSPS KAKENHI Grant Numbers 26291010 and 15H01463. O.N. is supported by the Basic Science and Platform Technology Program for Innovative Biological Medicine from the Japan Agency for Medical Research and Development, AMED, and the Council for Science, Technology and Innovation (CSTI), Cross-ministerial Strategic Innovation Promotion Program (SIP), "Technologies for creating next-generation agriculture, forestry and fisheries" (funding agency: Bio-oriented Technology Research Advancement Institution, NARO).

## Author contributions

S.H. solved the crystal structure and performed in vitro cleavage experiments with assistance from H.N.; J.S.G., O.O.A. and F.Z. performed the PAM identification assay and the indel analysis in human cells; T.H. and I.H. performed the indel analysis in mouse zygotes; R.I. assisted with the structural determination; S.H., H.N. and O.N. wrote the manuscript with help from all authors; H.N. and O.N. supervised all of the research.

## Additional information

**Competing interests:** J.S.G., O.O.A. and F.Z. are co-founders of Sherlock Biosciences. O.O.A. and J.S.G. are advisors for Beam Therapeutics. J.S.G. is a campus advisor of Benchling, Inc. F.Z. is a co-founder and advisor of Editas Medicine, Arbor Biotechnologies, Beam Therapeutics, and Pairwise Plants. H.N. is an advisor of EdiGENE. O.N. is a co-founder and advisor of EdiGENE. The remaining authors declare no competing interests.

