## [Peer Review File · Nature Communications]

Reviewers' comments:

Reviewer #1 (Remarks to the Author):

Hirano et al. report the activity and structure of Cas9 from *Corynebacterium diphtheriae*. Cas9 from *Streptococcus pyogenes* is widely used for programmable genome editing, and various other orthologs of Cas9 have been studied and applied to expand the number of genome editing tools. In particular, Cas9 orthologs that recognize different PAM sequences are desirable, as they could greatly expand the number of targets available to researchers in any given genome editing experiment. CdCas9 has previously been reported to recognize an A-rich PAM, which is substantially different from the G-rich PAMs observed for most other Cas9 orthologs. However, previous studies of CdCas9 suggested it had low activity for DNA cutting in vitro, and no genome editing activity in vivo. Here, the authors perform more in-depth activity studies of CdCas9, determining that the enzyme requires a slightly longer guide RNA strand (22 nt instead of 20 nt) for optimal activity. They characterize the PAM preferences in more detail, and demonstrate genome editing activity in mouse embryos, although the activity is very limited. The authors solve the structure of CdCas9 in complex with guide RNA and target dsDNA, which reveals the mechanism of PAM recognition for this ortholog. They find that interactions with the PAM are mediated through hydrogen bonding and van der Waals interactions that read out both the target and non-target strand of the PAM.

The structure presented in this manuscript adds to a growing catalog of Cas9 orthologs that could increase the sequence space targeted for genome editing. However, the low level of genome editing activity observed with this ortholog suggests that it will take significantly more development for CdCas9 to become a versatile tool on par with SpCas9. The authors provide some hypotheses about why the in vivo editing activity is low (e.g. guide RNA may be truncated to an inactive length), which may be helpful for further developing CdCas9 into a genome editing tool. Given the substantial interest in Cas9, this work will likely be of interest to researchers in the field of genome editing and CRISPR biology. I have a few concerns about the conclusions drawn from some of the data that should be addressed by the authors.

1. There are several instances in the text where the authors state that activity is “comparable” between different Cas9 variants. For example, the authors state that the graph shown in Figure 1C reveals comparable activity between CdCas9 and SpCas9. However, aside from the 30 s time point, the fraction cleaved is saturated at all time points in the experiment, meaning this experiment is not really a good comparison of the cleavage rates of the two proteins. At 30 s, it appears that CdCas9 has substantially less product than SpCas9. This experiment would be better if performed with shorter time points that fall within the exponential rise of percent cleaved, which could reveal a significantly lower cleavage rate for CdCas9 in comparison to SpCas9 and could partially explain the lower activity for in vivo genome editing.

2. Similarly, experiments shown in Figure 4D were performed at only a single 30 min time point, which is >10x more time than is necessary for complete cleavage at WT rates. This means the mutants tested could have up to 10x defects in cleavage that are masked by the long time point. In particular, the R1042A mutant is interesting because there appears to be a slight favorability for G at the second position of the PAM, based on the logo shown in Figure 1D. Do shorter time points reveal a slight cleavage defect for R1042A, reflecting the slight favorability for the interacting G? Is it possible that there are additive effects of having sub-optimal nucleotides at each position? For example, if position 2 is not a G, perhaps some nucleotides that are normally tolerated at positions 3-7 become sub-optimal.

3. The authors tested several constructs with varied PAMs for cleavage by CdCas9 (Figure 1E). However, they did not test the PAM sequences of the in vivo targets used for genome editing shown in Figure S1B and Table S2. The authors could test the two PAM variants shown in Figure S1B to ensure that cleavage of targets containing these PAMs is similar to the other PAMs tested. This could rule out

the possibility that CdCas9 may not be able to cut the in vivo target sites due to PAM identity. If some combinations of tolerated nucleotides are sub-optimal, as suggested in comment #2, it is possible that these PAMs may have lower activity.

4. The authors discuss the idea of allosteric control of cleavage (as observed for SpCas9 in Chen et al. and Dagdas et al. 2017) briefly in the discussion section, and this is indeed a possible explanation for why the 22-nt sgRNA is more active. The authors cite evidence that the sgRNA may be "processed" to 20-nt in human cells as a possible explanation for the low in vivo editing activity. If processing occurs, it is likely due to cleavage by non-specific cellular RNases, which may cleave any part of the sgRNA that is not protected by Cas9. This would likely happen in any type of cell, including the native host. Do the authors have any explanation for how the 22-nt crRNA would be maintained in the native organism?

Minor concern:

1. What sgRNA was used for the crystal structure (e.g. 20-nt or 22-nt)?

2. Lines 114-116: The authors state that they tested "27 plasmid targets, which contain N at each position in the GGGAAAC PAM." This wording is confusing, because it sounds like there is an N at each position (e.g. NNNNNNN) rather than that each potential nucleotide was tested, one-by-one, at position of the PAM. Also, only 21 of the plasmids tested would fit this description, as the other 6 are testing N's at multiple positions. This sentence could be reworded to clarify.

Reviewer #2 (Remarks to the Author):

Hirano et al. reports structure and function studies of *Corynebacterium diphtheriae* Cas9. The structure, although containing a partial CdCas9, sheds light on how CdCas9 recognizes its unique PAM sequence. The accompanied biochemical and in vivo analysis significantly elevates the structural work. The manuscript is clearly written and the quality of each experiment is high. I only have some minor concerns.

1. Authors clearly showed that the wild-type CdCas9 favors a longer sgRNA:DNA duplex (22mer) than that of SpCas9 in DNA cleavage. Can authors show if the DHH-CdCas9 behaves similarly in guide-dependent nicking DNA, i.e. requiring 22-nt guide? If, as authors pointed out, the PAM-distal region plays a role in activating the HNH domain, perhaps the DHH-CdCas9 is inactive (guide-dependent nicking) with the 20-nt guide?

2. If authors show that DHH-CdCas9 does not nick DNA with the 20-nt guide, then the structure presented represents an inactive complex. Authors may need to attempt crystallizing a 22-nt complex. It would be very satisfying to be able to compare the structures between the 20-nt and the 22-nt complex in order to identifying any possible conformational changes;

3. Even though authors have published a number of similar works on Cas9 with high quality figures, it still is helpful to describe all elements in the figures. Not all colors are described in some figures or their captions (for instance, Figure S2);

4. In the Method Details sample preparation section, authors neglected to include buffers compositions.

Reviewer #1:

Hirano et al. report the activity and structure of Cas9 from Corynebacterium diphtheriae. Cas9 from Streptococcus pyogenes is widely used for programmable genome editing, and various other orthologs of Cas9 have been studied and applied to expand the number of genome editing tools. In particular, Cas9 orthologs that recognize different PAM sequences are desirable, as they could greatly expand the number of targets available to researchers in any given genome editing experiment. CdCas9 has previously been reported to recognize an A-rich PAM, which is substantially different from the G-rich PAMs observed for most other Cas9 orthologs. However, previous studies of CdCas9 suggested it had low activity for DNA cutting in vitro, and no genome editing activity in vivo. Here, the authors perform more in-depth activity studies of CdCas9, determining that the enzyme requires a slightly longer guide RNA strand (22 nt instead of 20 nt) for optimal activity. They characterize the PAM preferences in more detail, and demonstrate genome editing activity in mouse embryos, although the activity is very limited. The authors solve the structure of CdCas9 in complex with guide RNA and target dsDNA, which reveals the mechanism of PAM recognition for this ortholog. They find that interactions with the PAM are mediated through hydrogen bonding and van der Waals interactions that read out both the target and non-target strand of the PAM.

The structure presented in this manuscript adds to a growing catalog of Cas9 orthologs that could increase the sequence space targeted for genome editing. However, the low level of genome editing activity observed with this ortholog suggests that it will take significantly more development for CdCas9 to become a versatile tool on par with SpCas9. The authors provide some hypotheses about why the in vivo editing activity is low (e.g. guide RNA may be truncated to an inactive length), which may be helpful for further developing CdCas9 into a genome editing tool. Given the substantial interest in Cas9, this work will likely be of interest to researchers in the field of genome editing and CRISPR biology. I have a few concerns about the conclusions drawn from some of the data that should be addressed by the authors.

1. There are several instances in the text where the authors state that activity is “comparable” between different Cas9 variants. For example, the authors state that the graph shown in Figure 1C reveals comparable activity between CdCas9 and SpCas9. However, aside from the 30 s time point, the fraction cleaved is saturated at all time points in the experiment, meaning this experiment is not really a good comparison of the cleavage rates of the two proteins. At 30 s, it appears that CdCas9 has substantially less product than SpCas9. This experiment would be better if performed with shorter time points that fall within the exponential rise of percent cleaved, which could reveal a significantly lower cleavage rate for CdCas9 in comparison to SpCas9 and could partially explain the lower activity for in vivo genome editing.

According to the reviewer's suggestion, we measured the cleavage activities of CdCas9 and SpCas9 in the time course including 0.25 min, and found that CdCas9-sgRNA22 almost completely cleaved the target DNA in 2 min, but its cleavage kinetics was slower than that of SpCas9 (Fig. 1b). We added the statement to the revised manuscript.

2. Similarly, experiments shown in Figure 4D were performed at only a single 30 min time point, which is >10x more time than is necessary for complete cleavage at WT rates. This means the mutants tested could have up to 10x defects in cleavage that are masked by the long time point. In particular, the R1042A mutant is interesting because there appears to be a slight favorability for G at the second position of the PAM, based on the logo shown in Figure 1D. Do shorter time points reveal a slight cleavage defect for R1042A, reflecting the slight favorability for the interacting G? Is it possible that there are additive effects of having sub-optimal nucleotides at each position? For example, if position 2 is not a G, perhaps some nucleotides that are normally tolerated at positions 3-7 become sub-optimal.

According to the reviewer's suggestion, we measured the cleavage activities of the CdCas9 mutants with sgRNA22 at 0.5 and 5 min, and found that the mutants including R1042A are less active than the wild-type protein (Fig. 6d). Furthermore, we measured the cleavage activities of the wild-type protein towards the GNGGAAC, GNGAGAC, and GNGAAGC targets. We found that CdCas9 cleaves the GGGGAAC/GGGAGAC/GGGAAGC targets more efficiently, as compared to the GHGGAAC/GHGAGAC/GHGAAGC targets (Fig. 6e), suggesting the functional importance of the interaction between Arg1042 and the second G nucleotide for the recognition of the suboptimal PAMs. We have added these data into the revised manuscript.

3. The authors tested several constructs with varied PAMs for cleavage by CdCas9 (Figure 1E). However, they did not test the PAM sequences of the in vivo targets used for genome editing shown in Figure S1B and Table S2. The authors could test the two PAM variants shown in Figure S1B to ensure that cleavage of targets containing these PAMs is similar to the other PAMs tested. This could rule out the possibility that CdCas9 may not be able to cut the in vivo target sites due to PAM identity. If some combinations of tolerated nucleotides are sub-optimal, as suggested in comment #2, it is possible that these PAMs may have lower activity.

According to the reviewer's suggestion, we measured the *in vitro* cleavage activities of CdCas9 towards the target plasmids with the GTATAAT and TGGTAAT PAMs, and confirmed that CdCas9 cleaves these targets *in vitro* (Supplementary Fig. 3b). We have added these data into the revised manuscript.

4. The authors discuss the idea of allosteric control of cleavage (as observed for SpCas9 in Chen et al. and Dagdas et al. 2017) briefly in the discussion section, and this is indeed a possible explanation for why the 22-nt sgRNA is more active. The authors cite evidence that the

sgRNA may be “processed” to 20-nt in human cells as a possible explanation for the low in vivo editing activity. If processing occurs, it is likely due to cleavage by non-specific cellular RNases, which may cleave any part of the sgRNA that is not protected by Cas9. This would likely happen in any type of cell, including the native host. Do the authors have any explanation for how the 22-nt crRNA would be maintained in the native organism?

As the reviewer pointed out, it is possible that the cleavage activity of CdCas9 is reduced due to the guide RNA trimming by endogenous RNases in native host cells. Nonetheless, a guide RNA with a 20-nt guide sequence can support CdCas9-mediated dsDNA cleavage, albeit at reduced efficiency, which may be sufficient for the destruction of foreign DNAs. Alternatively, given that CdCas9 also cleaves ssDNA targets with a 20-nt guide RNA (Ma *et al. Mol. Cell* 2015), the ssDNA cleavage activity may be sufficient to degrade foreign ssDNAs (*e.g.*, M13 phage). Thus, CdCas9 may function in the CRISPR-Cas immune system, although CdCas9 failed to robustly mediate genome editing in human cells.

Minor concern:

1. What sgRNA was used for the crystal structure (e.g. 20-nt or 22-nt)?

We used a 20-nt guide sgRNA for crystallization. We described the guide length of sgRNA in the revised manuscript.

2. Lines 114-116: The authors state that they tested “27 plasmid targets, which contain N at each position in the GGGAAAC PAM.” This wording is confusing, because it sounds like there is an N at each position (e.g. NNNNNNN) rather than that each potential nucleotide was tested, one-by-one, at position of the PAM. Also, only 21 of the plasmids tested would fit this description, as the other 6 are testing N’s at multiple positions. This sentence could be reworded to clarify.

According to the reviewer’s suggestion, we have rewritten the text.

Reviewer #2:

Hirano et al. reports structure and function studies of Corynebacterium diphtheriae Cas9. The structure, although containing a partial CdCas9, sheds light on how CdCas9 recognizes its unique PAM sequence. The accompanied biochemical and in vivo analysis significantly elevates the structural work. The manuscript is clearly written and the quality of each experiment is high. I only have some minor concerns.

1. Authors clearly showed that the wild-type CdCas9 favors a longer sgRNA:DNA duplex (22mer) than that of SpCas9 in DNA cleavage. Can authors show if the DHNH-CdCas9 behaves similarly in guide-dependent nicking DNA, i.e. requiring 22-nt guide? If, as authors

pointed out, the PAM-distal region plays a role in activating the HNH domain, perhaps the DHNH-CdCas9 is inactive (guide-dependent nicking) with the 20-nt guide?

According to the reviewer's suggestion, we measured the target strand cleavage activity of the HNH domain, using the D10A RuvC-inactive mutant and a 20- or 22-nt guide sgRNA. The CdCas9 D10A mutant exhibited efficient cleavage activity with a 22-nt guide rather than a 20-nt guide, indicating the importance of the PAM-distal RNA-DNA base pairing (21–22 bp) for the HNH domain activation in CdCas9 (Supplementary Fig. 1c). We have added these data into the revised manuscript.

2. If authors show that DHNH-CdCas9 does not nick DNA with the 20-nt guide, then the structure presented represents an inactive complex. Authors may need to attempt crystallizing a 22-nt complex. It would be very satisfying to be able to compare the structures between the 20-nt and the 22-nt complex in order to identifying any possible conformational changes.

According to the reviewer's suggestion, we performed crystallization experiments, using the CdCas9- Δ HNH variant, a 22-nt guide sgRNA, and a target DNA. However, we failed to obtain crystals under the similar crystallization condition. It will be important to determine the structure of full-length CdCas9 with a 22-nt guide RNA to elucidate the activation mechanism in future studies.

3. Even though authors have published a number of similar works on Cas9 with high quality figures, it still is helpful to describe all elements in the figures. Not all colors are described in some figures or their captions (for instance, Figure S2).

According to the reviewer's suggestion, we labeled all elements of Cas9, sgRNA, and target DNA in figures.

4. In the Method Details sample preparation section, authors neglected to include buffers compositions.

According to the reviewer's suggestion, we described the buffer compositions in the procedure for the protein purification.

REVIEWERS' COMMENTS:

Reviewer #1 (Remarks to the Author):

The authors have addressed all of my prior concerns. The new data supports their conclusion and the manuscript is now ready for publication.

Reviewer #2 (Remarks to the Author):

Authors made sufficient efforts in addressing the comments I raised. However, I noticed an inconsistency that requires authors' further explanation. Although in vitro cleavage experiment showed that the sgRNA22 is the most efficient in cleaving DNA, sgRNA24 produced substantially more indels than sgRNA22 in mouse zygotes for Tet1EX12 (supplementary Table 1). Can authors check their data or provide an explanation of this inconsistency?

Responses to the reviewers' comments

Reviewer #1:

The authors have addressed all of my prior concerns. The new data supports their conclusion and the manuscript is now ready for publication.

We thank the reviewer for the agreement with the publication.

Reviewer #2:

Authors made sufficient efforts in addressing the comments I raised. However, I noticed an inconsistency that requires authors' further explanation. Although in vitro cleavage experiment showed that the sgRNA22 is the most efficient in cleaving DNA, sgRNA24 produced substantially more indels than sgRNA22 in mouse zygotes for Tet1EX12 (supplementary Table 1). Can authors check their data or provide an explanation of this inconsistency?

We checked our data of the indel analysis in mouse zygotes, and confirmed that sgRNA24 (56%) induced indels at the *Tet1EX12* target site more efficiently as compared to sgRNA22 (8%). One possible reason may be the processing at the 5'-terminal region of the sgRNAs in cells. Nonetheless, sgRNA22 (4.8%) induced slightly more indels at the *Tet1EX4* target site as compared to sgRNA24 (0%). Thus, further studies will be needed to fully elucidate the optimal guide lengths of CdCas9 for *in vivo* genome editing.

We thank you very much for your kind consideration again.